# Community composition exceeds area as a predictor of long-term conservation value

**Jacob D. O'Sullivan** \*, **J. Christopher D. Terry**, **Ramesh Wilson**, **Axel G. Rossberg**

School of Biological and Behavioural Sciences, Queen Mary University of London, London, United Kingdom

\* j.osullivan@zoho.com

## Abstract

Conserving biodiversity often requires deciding which sites to prioritise for protection. Predicting the impact of habitat loss is a major challenge, however, since impacts can be distant from the perturbation in both space and time. Here we study the long-term impacts of habitat loss in a mechanistic metacommunity model. We find that site area is a poor predictor of long-term, regional-scale extinctions following localised perturbation. Knowledge of the compositional distinctness (average between-site Bray-Curtis dissimilarity) of the removed community can markedly improve the prediction of impacts on regional assemblages, even when biotic responses play out at substantial spatial or temporal distance from the initial perturbation. Fitting the model to two empirical datasets, we show that this conclusions holds in the empirically relevant parameter range. Our results robustly demonstrate that site area alone is not sufficient to gauge conservation priorities; analysis of compositional distinctness permits improved prioritisation at low cost.

**Data Availability Statement:** Representative subset of simulation results uploaded to 10.6084/ m9.figshare.21731702.

## Author summary

Species are being driven extinct at a rate 1, 000–10, 000 times greater than natural background due to a suite of anthropogenic pressures on ecosystems. Conservation of biodiversity frequently involves localised interventions to protect communities of species from such anthropogenic impacts. Due to the complexity and adaptability of ecosystems, understanding the relationship between local impacts and regional-scale patterns in biodiversity is a major challenge. We use a computational model of a complex ecological assemblage, previously shown to reproduce a variety of key macroecological patterns, to study the long-term impacts of localised perturbation on regional scale ecosystems. Using this experimental approach, we characterise the 'conservation value' of different locations by reference to the loss of species at the regional scale following local site destruction. We find that, while both the area of the impacted site and the compositional rarity of the community it supports are important predictors of long-term conservation value, composition is a more important predictor. In view of this result, we argue that local composition should be given greater weight when assessing conservation priorities and designing conservation and management programmes.

**Funding:** This work forms part of the project 'Mechanisms and prediction of large-scale ecological responses to environmental change' with funds awarded to AGR by the Natural Environment Research Council (NE/T003510/1). JDO, JCDT and AGR received salaries from this grant. The funders had no role in study design, data collection and analysis, decision to publish, or preparation of the manuscript.

**Competing interests:** The authors have declared that no competing interests exist.

## Introduction

Habitat loss due to conversion of natural landscapes is the leading cause of biodiversity loss today [1, 2]. Immediate species losses that result when the habitats of endemic species are destroyed only represent part of the impact of land conversion. Additional losses may occur as 'extinction debts' are paid [3]. These additional extinctions can arise due to a suite of complex processes that ripple through the wider landscape, often involving multiple species [4–9]. The complexity of these ecological responses to habitat loss makes predicting the long-term outcomes of localised impacts a major challenge. Essential to meeting this challenge is an understanding of how changes in the abiotic and biotic structure of the landscape are likely to interact [10].

Decisions in conservation ecology often require identification of the least worst outcomes of landscape conversion [11]. For such assessments, sufficient mechanistic understanding of the biodiversity impacts of habitat loss is required [12]. Predictions of such impacts often rely on phenomenological models of species-area relationships (SAR) [12–14], which assume that impacts follow simple scaling relationships (e.g. [15, 16]). However, the scaling of diversity with area arises due the fact that ecological assemblages are internally heterogeneous: it is usually not area *per se* that determines species richness but the diversity of ecological associations that a landscape can support. As such, it is plausible that metrics that directly quantify the internal diversity structure of an assemblage may outperform area alone as predictors of a site's contribution to regional diversity.

Here, we explore the long-term outcomes of habitat destruction using a spatially explicit simulation model called the Lotka-Volterra Metacommunity Model (LVMCM), which has been shown to reproduce a large variety of well-documented macroecological patterns including a strongly right-skewed range size distribution, a power-law SAR and related species time relation, and the time invariance of key macroecological structures despite continuous turnover in species composition [17, 18]. The present study employs this model system to ask applied questions about the impact of perturbation on regional biota.

We model habitat conversion as the complete removal of sites from a metacommunity, study the impacts of those removals after simulating the metacommunity response, and find that indeed the biotic impacts following site removal can be complex. Secondary extinctions including extinction cascades are common. These cascades can cause extinctions of populations distant from the removed site. The area of the removed site only weakly correlates with *conservation value*, which we define as the proportional loss of species at the regional scale after a long relaxation. A stronger predictor of regional extinctions is often the compositional distinctness of the removed community, despite the cascading, far-reaching impacts removal can have. To test whether empirical systems fall into the parameter space where compositional distinctness is a stronger predictor than area, we developed a method to fit the LVMCM to biodiversity patterns derived from empirical species-by-site tables. Using this method, we demonstrate not only the higher predictive power of compositional distinctness for empirical systems, but also the future potential for mechanistic metacommunity models as decision support tools in applied conservation [10].

## Summary of methods

### Model overview

Lotka-Volterra models with additional terms describing dispersal have been studied in various contexts (e.g. [19–21]). Here we forgo common model validation analyses which have been a repeated focus of previous work and instead use this tried and tested approach to ask applied

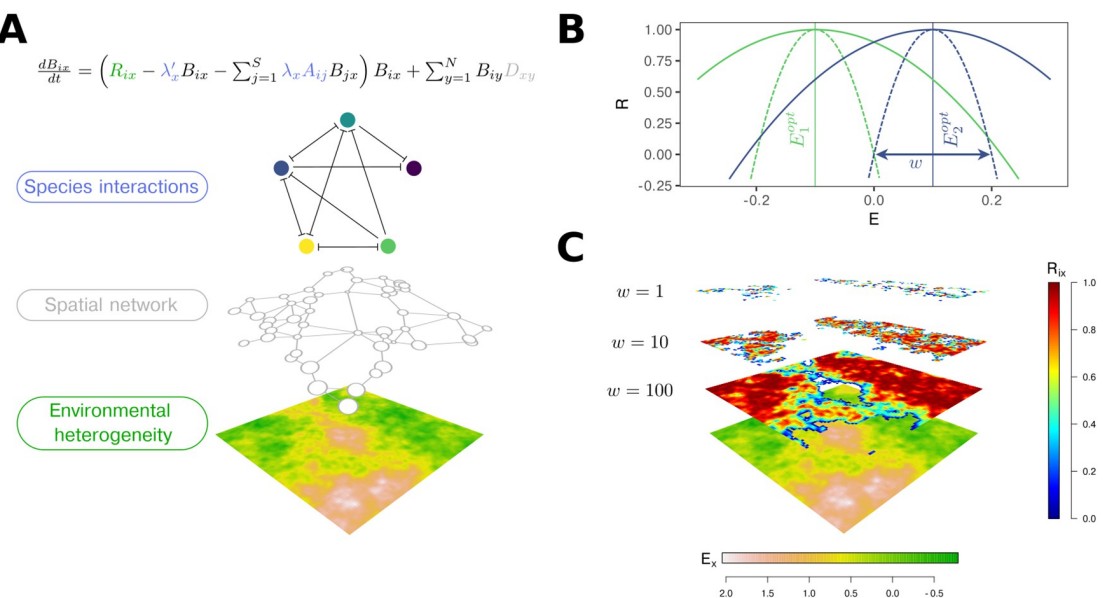

**Fig 1. Elements of the metacommunity model. A**: Temporal dynamics in local biomass ($B_{ix}$) are modelled as functions of local intrinsic growth rates ($R_{ix}$) mediated by competitive interactions ($A_{ij}B_{jx}$) and immigration pressure, a function of the distance from adjacent sites ($B_{iy}D_{xy}$). The abiotic environment is represented by a spatially autocorrelated distribution of at least one variable. The spatial network is a random planar graph in which local sites of unequal area are modelled by scaling the local interaction coefficients by $\lambda_x$ and $\lambda'_x$ (see Section A in S1 Text for details). **B**: Intrinsic growth rates $R = R_{ix}$, representing species' Hutchinsonian niches, are modelled as quadratic functions of the environmental variables. Niche width is controlled by a parameter $w$ and each species is assigned a randomly sampled environmental optimum $E_i^{\mathrm{opt}}$ (solid lines exemplify large $w$, dashed lines small $w$, colours represent different species). **C**: Three examples illustrating of how niche width $w$ affects the distribution of areas of positive $R$ over the landscape, highlighting the relationship between niche width and effective heterogeneity.

questions with direct relevance to conservation biology. The LVMCM [17, 18] extends the conventional Lotka-Volterra competition model to a spatially explicit system of connected sites (Fig 1). It models the dynamics of a metacommunity formed by a guild of competing species. Sites differ in their suitability for each species and their total area (see Section A in S1 Text for details on scaling local site area), and nearby communities are connected by dispersal. The underlying environment is modelled as a spatially autocorrelated random field with mean zero and unit standard deviation. Species responses to the environment are modelled by quadratic environmental response functions, with each species allocated a random environmental optimum. Species disperse between sites according to an exponential dispersal function. We kept the width of the environmental response function, $w$, and the characteristic dispersal length $\ell$ fixed for all species in a given simulation, but varied them systematically between simulations. For this study we kept the number of sites fixed at 20 while, under variation of these key parameters, the regional $\gamma$-diversity ranged from 250 to 2500 species. In this way we generated a large and heterogeneous set of simulated metacommunities from which we aim to infer general relationships between long-term impacts and the area or compositional distinctness of the removed site.

We allowed metacommunity models to self-organise via a regional assembly process [22] through which species invade the metacommunity and distribute across the landscape, responding to the abiotic and biotic conditions in the sites. We then systematically perturbed assembled metacommunities by removing each site in turn, simulating the biotic response and measuring long-term impacts of single site removal at the regional scale. Simulated perturbation experiments are of great value in this context since a thorough empirical test would

involve systematic removal of sites and extinctions of species in the real world. Such an approach would be a) highly unethical (though not entirely unheard of, [23]) and b) require many decades to perform due to the need to start each new experiment from a 'healthy' meta-community. More details on the assembly and perturbation procedures are given in the Methods.

## Predicting long-term species losses

Process-based metacommunity models like the LVMCM permit direct comparison between the immediate effects and long-term outcomes of perturbations. Immediate species losses can be predicted by asking which species have a global range limited to the removed site. We denote the predicted immediate species loss $\Delta\gamma_0$. In contrast, we denote by $\Delta\gamma$ the actual long-term species loss determined by simulating metacommunity dynamics. We define the conservation value of the removed site as the proportional long-term species loss, relative to the pre-disturbance regional species richness $\gamma^*$.

We first asked how $\Delta\gamma/\gamma^*$ depends on model parameters. Determining these parameters empirically, however, can be costly. Therefore we also tested how well $\Delta\gamma/\gamma^*$ can be predicted by the proportion of biomass immediately removed, $B_x^*/B^*$, where $B_x^*$ and $B^*$ represent the pre-disturbance biomass of the removed site $x$ and of the entire metacommunity, respectively. We use $B_x^*$ as a proxy for the area of site $x$, with the advantage that $B_x^*$ is directly represented in LVMCM model communities. As a second predictor we use a measure of compositional distinctness, $\bar{\beta}_x$, defined as the mean Bray-Curtis [24] dissimilarity of the focal site $x$ to all other sites.

We assessed the effects of $B_x^*/B^*$, $\bar{\beta}_x$ and various spatial network properties (centrality measures) on proportional species losses $\Delta\gamma/\gamma^*$ using simple multivariate linear regression, with predictors scaled to mean zero and unit variance and the best model being selected by comparison of AIC [25]. Goodness-of-fit of predictive models was assessed using the adjusted $R^2$. The proportions of variance explained by area and compositional distinctness were then partitioned using partial regression redundancy analysis [26].

## Fitting the LVMCM to empirical data

We demonstrate how the LVMCM can be fit to data using two datasets, one describing the distribution of diatoms (D) in lakes straddling the Peruvian-Bolivian border [27], the other the distribution of Lichen-Fungi (L) in Eastern Brazil [28]. These datasets were chosen for their high species richness and because we can assume that the ecological interaction networks of these guilds are reasonably represented by competition within a single trophic level. For dataset D, which covers a large region, we selected a subset of the full database for which the first two principal components of the key environmental variables formed a distinct cluster, including 19 sites and 221 taxa (Fig 2A). For dataset L we reduced the total number of sites (and the computational effort) by pooling observations separated by less than 20km. This left 12 sites and 784 taxa (Fig 2B). For this dataset, environmental variables were extracted from publicly available remote sensing databases.

In Sections B and C in S1 Text we give a detailed summary of the model fitting procedure that we outline here. The spatial structure (spatial distance matrix) and key environmental variables from the empirical dataset were input directly into the model, defining the distance between sites and the underlying abiotic environment. Additionally, the observed richness of local sites, $\alpha_{\mathrm{obs}}$ was used to define the effective area of the sites in the corresponding simulation model. We then used a battery of simulations to estimate the abiotic niche width parameter $w$ that best fit the empirical species by site table. This was done by quantitatively comparing

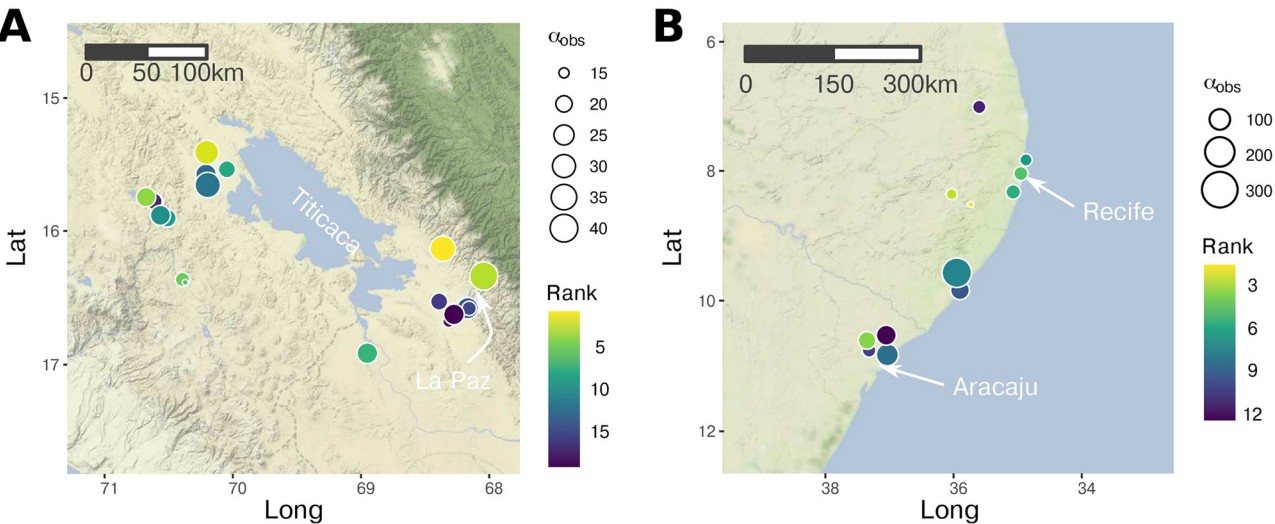

**Fig 2. Datasets used to fit the LVMCM. A**: The subset of the Andean diatom database [27] straddling Lake Titicaca and the Peruvian-Bolivian border. **B**: The Brazilian lichen-fungi database [28] with nearby samples pooled together. The colour of the points represents the ranked impact of site removal on regional diversity in *in silico* removal experiments, with the lightest colour corresponding to the site with the greatest impact. The size of the points represents the observed species richness $\alpha_{obs}$. Map tiles by Stamen Design, under CC BY 3.0. Data by OpenStreetMap, under ODbL.

the occupancy frequency distribution (OFD) [29] for the empirical observation to that of the simulation under systematic variation of *w*.

The LVMCM is a parameter rich model, making direct fitting to data impossible. Therefore we developed a simplified patch occupancy model that predicts high-level metacommunity properties with a single fitting parameter, which we termed the 'mixing rate' *m*. By construction this mixing rate defines the ratio between the mean rate at which sites are colonised by populations in adjacent communities and the rate of invasion into the assemblage from outside of the metacommunity. While unsuitable for in-depth analyses of biotic responses to perturbation, the simplicity of the patch occupancy model means it can be directly fitted to both empirical observations and LVMCM model communities, thus serving as a bridge between the complex LVMCM simulations and empirical ecosystems. Using an intermediate model to fit the LVMCM to observation data in this way is an unconventional procedure, but gives a reasonable indication of the approximate location in the model's parameter space that best corresponds to the real-world assemblage. The theory behind this simple model and the procedure for model fitting are summarised in Section D in S1 Text.

The parameter *m* can be estimated from the OFD for a given dataset. OFD generally offer little scope for estimating niche widths since they include no information of spatial structure or environmental heterogeneity. Here, however, we use LVMCM models with spatial and environmental distributions matching the empirical observation and therefore are in a position to approximate typical niche widths from OFD. For LVMCM models with underlying heterogeneity taken from the empirical observation, the abiotic niche width parameter space of the model was scanned. The value of *w* that best reproduces the mixing rate of the empirical OFD was then located. This value gives a meaningful estimate of the typical niche widths relative to the total environmental variation in the assemblage that, importantly, includes some explicit consideration of the impact of biotic interactions on species ranges. Note that, should more complex dispersal dynamics or ecological interaction types be included in the model, these might additionally impact the shape of the OFD and therefore require parameter

estimation. Such complexity is absent from the framework used here and therefore we fit the LVMCM to the empirical OFD by varying $w$ alone.

## Results and discussion

Even though none of the sites removed in simulation experiments comprised more than 12% of the total regional biomass/area, we detected regional extinction of at least one species in over 75% of cases. The highest impact of a single site removal was a proportional loss of $\Delta\gamma/\gamma^* = 0.23$.

### The dynamics of extinction following habitat removal

Proportional long-term species loss $\Delta\gamma/\gamma^*$ decreased with increasing abiotic niche width (Fig 3A). This is plausible, since with wider niches single-site removal tends to remove a smaller proportion of the area representing a species' Hutchinsonian niche. By contrast we found that dispersal length had surprisingly little effect on $\Delta\gamma/\gamma^*$ (Fig 3A).

The process by which species are lost following site removals was often complex. In Fig 3B we show a random sample of these complex extinction dynamics for various dispersal lengths with abiotic niche width fixed at $w = 0.63$, the value for which impacts were greatest. We find that site removals can either lead to the loss of endemics only (cummulative extinctions effectively independent of time lag), or trigger extinction cascades, which can play out over long times. These extinction cascades, particularly prevalent in higher dispersing metacommunities, demonstrate the complexity of potential metacommunity responses to site removal.

The secondary extinctions of non-endemics can be categorised into two distinct types. Those occurring due the disruption of mass effects (sink populations which are lost following the removal of source sites), and those occurring due to a complex restructuring of the metacommunity as species ranges shift in space. Extinctions of the first type typically occur within around 50 unit times and in sites adjacent to the removed site. The average distance between the site in which a species was last detected and the original perturbation was 0.05$L$, where $L$ is the side length of the model landscape, in the first 50 unit times (smallest niche width, across

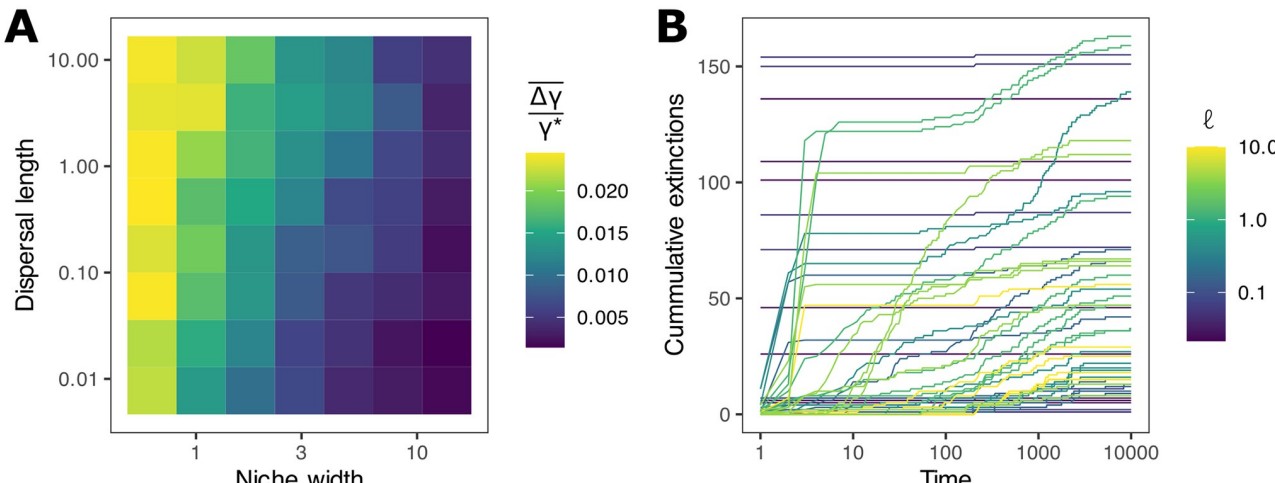

**Fig 3. Species extinctions following site removal. A**: The average across sites and replicate simulation models of the proportional species losses $\Delta\gamma/\gamma^*$ for all combinations of the abiotic niche width and dispersal length. **B**: Documentation of secondary extinctions resulting from the interruption of source-sink dynamics and extinction cascades. Here we show the outcome of removing sites for the smallest niche width $w = 0.63$. When dispersal lengths $\ell$ are particularly short and mass effects weak, secondary extinctions due to spatial restructuring do not occur.

dispersal lengths). The second type can occur much later and at almost any site in the meta-community. For extinctions occurring after more than 50 unit times, the average distance of the final local extinction in a given species' decline to extinction was $0.45L$. For species lost after 120 unit times, the distance from the initial perturbation had a mean $0.5004L$, statistically indistinguishable from the average inter-site distance $0.5002L$, suggesting the location of subsequent species losses was essentially random with respect to the initial impact.

## Predicting long-term impacts based on empirically measurable quantities

Predicting the outcome of single site removals, given the complexity of the biotic response, is non-trivial. It is not the case that species losses in the model can be generically explained by reference to immediate losses of endemics. Instead, we test whether area and compositional distinctness, both empirically accessible properties of communities, correlate with the long-term outcomes of the often complex structural redistribution precipitated by single site removal.

We find a clear positive association between the conservation value of sites and $B_x^*/B^*$, but with substantial spread (Fig 4, Spearman's $\rho = 0.33$, all parameter combinations pooled). In contrast, long-term species losses were more strongly associated with compositional distinctness of the removed site $\bar{\beta}_x^s$ (Fig 4, $\rho = 0.52$, all parameter combinations pooled).

The best model selected by AIC included both proportional area removed and compositional distinctness of removed site as predictors of long-term species losses. Surprisingly, none of the standard centrality measures available (degree-, closeness-, betweenness-, eigenvector centrality) were selected as predictors.

Decomposing the simulation results by $w$, we find that the strength of the association between long-term impacts and the properties of the removed site varies considerably with nice width. In Fig 5 we show how the $R^2$ and the regression coefficients of the multivariate linear models vary over the parameter space.

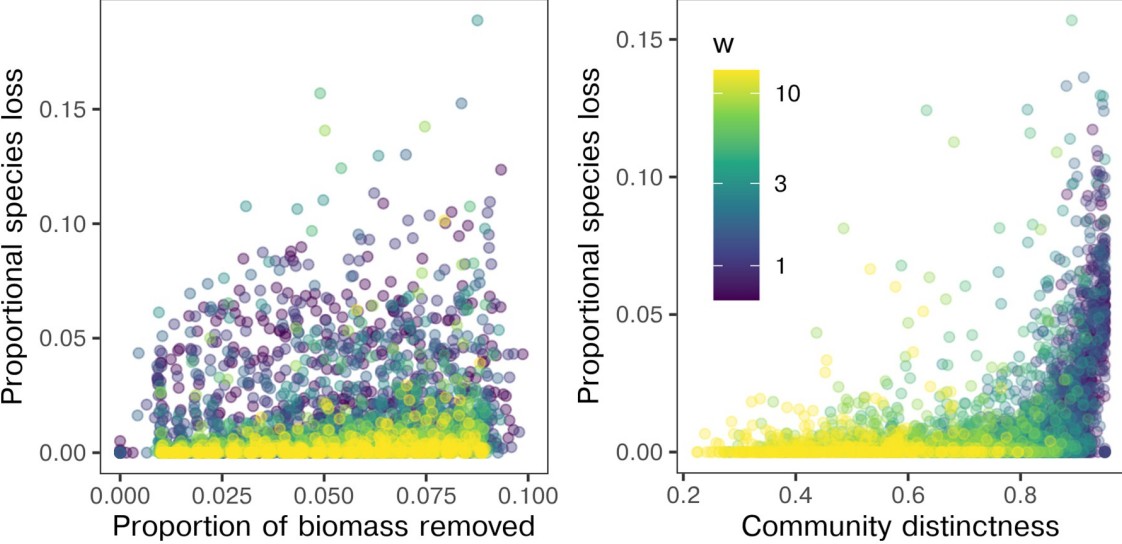

**Fig 4. Predicting long-term species losses.** The proportional long-term species losses following site removal plotted against the proportion of biomass removed initially, a proxy for site area, and the compositional distinctness of the removed site. Here we show a random subset of the simulated removal experiments for visual clarity. Dispersal length had little impact on long-term outcomes. In contrast, proportional losses were greatest for small niche widths $w$ largely due to the self-organised link between niche width and local community distinctness.

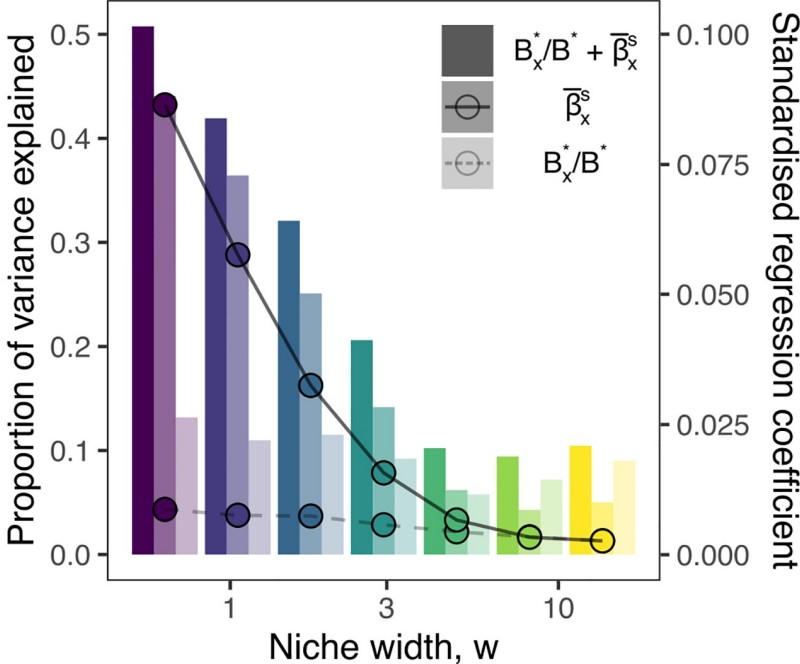

**Fig 5. Dependence of predictive power on niche width.** We find a clear decay with increasing niche width $w$ in the variance explained (bars) by both the full model (combining both predictors $B_x^*/B^*$ and $\overline{\beta}_x$) and the model which includes only distinctness ($\overline{\beta}_x$). In contrast, the proportion of variance explained by area ($B_x^*/B^*$) was largely independent of $w$. Coincident with the decay in variance explained, the standardised regression coefficients (points) also decayed with niche width. The standardised effect of area (dashed line) was consistently lower than that of compositional distinctness (solid line).

For the smallest niche widths studied here, the $R^2$ (bars in Fig 5) of the full model (both predictors) increased up to 0.55 for the smallest niche width. For large niche widths $R^2$ stabilised at 0.13. Thus, predicting $\Delta\gamma/\gamma^*$ when niche widths are large relative to environmental variation is particularly challenging. This is partly due to the fact that long-term species losses are rare when environmental filtering is weak, but may also reflect the fact that the biotic response to perturbation depends much more on dispersal and local competitive effects—absent from the multivariate regression—where environmental effects are largely neutral.

The proportion of variance explained by area following partial regression redundancy analysis was largely consistent across niche width parameterisations. In contrast, the proportion of variance explained by compositional distinctness decayed with increasing niche width. As a result, for the largest niche widths studied, the variance explained by area actually exceeds that by composition, though only once total $R^2$ had dropped to its minimum. The standardised regression coefficients estimated after scaling the predictors to mean zero and unit variance (points in Fig 5) show that for all but the widest abiotic niches, the effect of compositional distinctness exceeded that of area on long-term losses (solid and dashed lines respectively).

By repeating this analysis for random sub-sets of the simulated metacommunities containing a fixed number of species, we verified that these differences are not due to differences in overall species richness of low- and high-$w$ metacommunities.

Thus, we conclude that compositional distinctness typically outperforms biomass as a predictor of long-term losses, despite the fact that the population scale impacts are not uniquely felt in or near the impacted site. In order to estimate the long-term effects of habitat

destruction on biodiversity, it is critically important to take the community composition of the affected areas into account.

### Fitting the LVMCM to empirical observation

In view of the strong parameter dependence of the predictive power of compositional distinctness, it is critical to identify which region of the parameter space is most representative of natural ecosystems at large scales. We constructed a set of LVMCM metacommunities with spatial, environmental and local species richness distributions taken from the two empirical datasets. Using the best fitting patch occupancy model as an intermediate between the empirical data and the LVMCM simulation models we found a quantitative match between the OFD of dataset D and the corresponding simulation when $w = 3.46$. For dataset L, $w = 1.08$ gave the closest match between the simulation and the observation.

The simulated OFD deviate from the empirical OFD in two important respects (Fig 6A). The empirical OFDs tend to have a sharper peaks at single site occupancy than the simulations and slightly wider right tails. This is most likely due to the model's simplifying assumption that all species have the same abiotic niche width. Inspection of the OFD for various niche widths (Fig D in S1 Text) suggests that a better fit to the empirical distribution could be achieved by including some interspecific differences in abiotic niche width. We did not attempt to incorporate such differences in our model to avoid over-parameterization.

Despite these differences, our simple fitting procedure generated model metacommunities with macroecological OFD approximately matching that of the empirical observation. The key finding is that both datasets are fit by abiotic niche widths of order $w \approx 1$—representing around one standard deviation of the landscape scale heterogeneity—suggesting the parameter space in which removal experiments predict compositional dissimilarity *substantially* exceeds biomass as a predictor of conservation value is the most biologically relevant.

The primary goal of this fitting procedure was an assessment of the empirically relevant parameter space. However, this indirect fitting procedure also gave us an opportunity to apply the same removal experiment to a simulated metacommunity with spatial, environmental, local richness and species occupancy properties matching real-world assemblages. From these simulated experiments using fitted models, we made a mechanistic assessment of the rank conservation priority of each site (Fig 2). Simulated metacommunities with macroecological characteristics matching empirical data sets offer a valuable arena within which to explore the impacts of perturbations regional biota, though we note that further work to refine this fitting procedure is needed.

In these experiments the proportional drop in regional diversity $\Delta\gamma/\gamma^*$ after removal of a single site ranged from 0.03 to 0.13 for dataset D and 0.004 to 0.15 for dataset L (Fig 6B). In both cases and in common with previous results on random metacommunities not fit to observations, $B_x^*/B^*$ was a rather poor predictor of simulated species losses. On the other hand, a strong non-linear relationship between long-term impact and $\bar{\beta}_x^s$ was found, consistent with the outcomes shown in Fig 5. For dataset D, conservation priority typically increased with site altitude as spatial isolation and deviation from regionally typical abiotic conditions increased. Surprisingly, for dataset L the greatest impacts occurred when either of the two smallest sites were removed. This apparently incongruous result is predominantly explained by the fact that for this dataset observed species richness was negatively associated with environmental rarity measured as the mean Euclidean distance of the focal site from all other sites in environment space (Spearman's $\rho = -0.56$, $p = 0.057$). Compositional distinctness typically correlates with environmental rarity, though additional metacommunity processes (biotic filtering, mass effects) can also account for a proportion of internal biotic heterogeneity [30]. Thus, the least

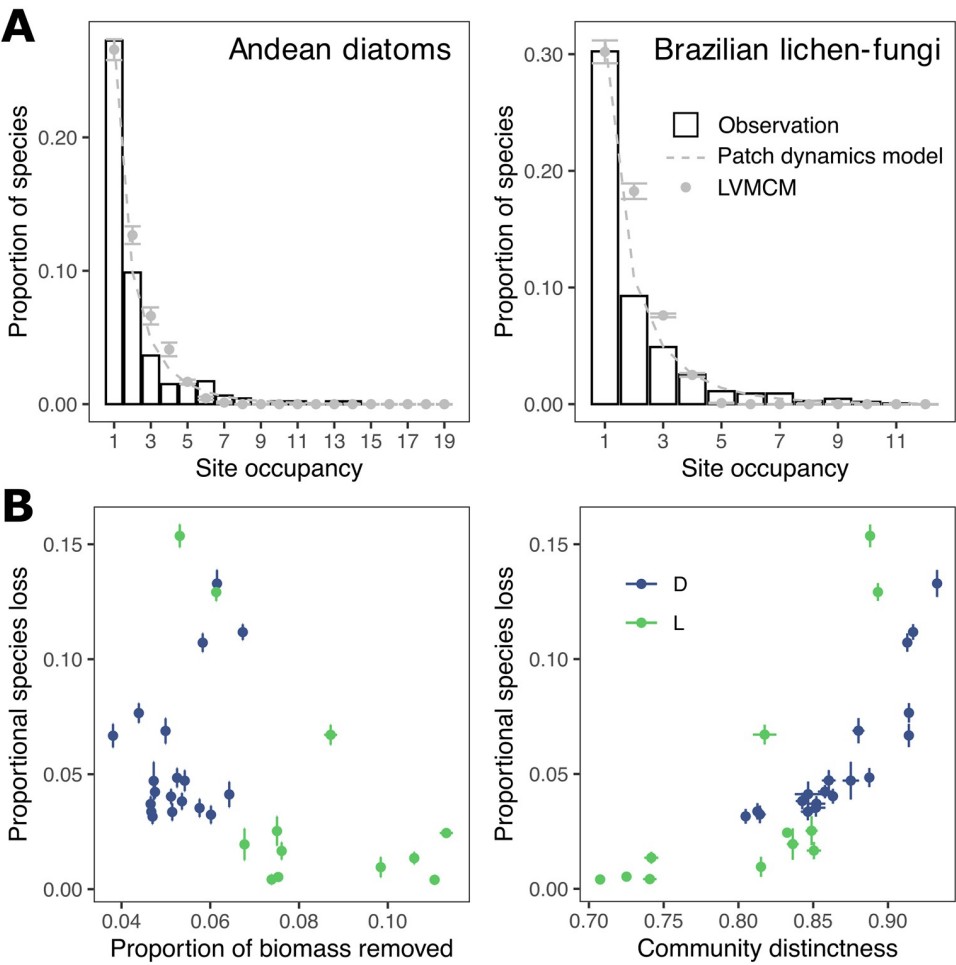

**Fig 6. Fitting the LVMCM to empirical OFD and simulated habitat loss. A**: Bars represent the empirical OFDs. The grey curves are the steady state OFD of the patch occupancy model with the mixing rate fitted to the observation. Grey points are the mean occupancies of 10 replicate LVMCM simulations with parameters that best fit the empirical observations. **B**: Having fitted the simulation to the broad macroecology of the empirical systems, we then performed simulated removal experiments on the best fitting models (OFDs of these shown with grey points in panel **A**). In **B** we show the outcomes of these removal experiments for simulated metacommunities which best match the Andean diatom (D, blue) and Brazilian lichen-fungi (L, green) datasets. Each point represents a site. In all panels, error bars represent standard deviation over 10 replicate assemblies.

species rich sites were also the most abiotically and biotically distinct and therefore of highest conservation value. For the remaining sites in the models matching the macroecology of dataset L, proportional species losses were largely independent of site area.

We acknowledge that testing the accuracy of these predictions is a challenge, however the present study demonstrates the possibility of bespoke, simulation-based assessment of conservation value, but leaves further refinements and validation of the procedure to future work.

## Conclusions

The biotic response of metacommunities to localised perturbation can be complex and far reaching. While endemic species are by definition lost immediately, secondary extinctions can be substantial as metacommunities restructure. Predicting the long-term impacts of this

restructuring is a major challenge, and one that currently can only be studied using process-based metacommunity models like the LVMCM. With this study we have shown that compositional distinctness can substantially exceed area alone as a predictor of long-term impacts, including secondary extinctions, on biodiversity following habitat destruction.

Compositional distinctness, measured in terms of average dissimilarity, is likely to be empirically accessible because Bray-Curtis dissimilarity between sites is numerically dominated by the locally dominatant populations, the sizes of which are readily quantified. However, the usefulness of the predictor is based not only on what it tells about the dominating species, but also what the dominating set of species tells about the abiotic characteristics of a site. The compositional distinctness of the dominant taxa is therefore likely to be informative of the distinctness of the non-dominant taxa, which is harder to measure.

While at first sight it might appear that application of this predictor requires a well-defined metacommunity to average over (which in practice may be ambiguous), this is not actually the case. Addition or omission of sites from a dataset affects ordering of two sites in terms of mean Bray-Curtis dissimilarity only for those other sites in the dataset that have compositions similar to either of the two sites.

To illustrate the relevance of our results, we note that the recently published first draft of the Post-2020 Global Biodiversity Framework [31] sets as a its primary goal the enhancement of ecosystem integrity, including increasing the area and connectivity of natural ecosystems, ensuring the robustness of populations and the reduction in extinction rates. Accompanying this document is a set of indicators proposed to help monitor progress toward these goals [32]. We can roughly categorise the 56 indicators applicable to the primary goal of the Framework into those relating to (i) area, extent or coverage, (ii) species-level assessments (of extinction risk, habitat integrity etc.) (iii) intactness (describing the proportion of historical assemblages that persists today) and (iv) spatial beta diversity intrinsic to an ecological unit (region, nation). The number of indicators in each category in the current draft are (i) 19, (ii) 12, (iii) 3 and (iv) 2. Thus, while our demonstration that compositional rarity is a key biotic quantity that must be conserved in order to protect regional biotas may be intuitive in hindsight, the weighting of area and composition in key management tools currently remains strongly skewed toward area-based measures. We therefore advocate for increased prioritisation of descriptors of the internal structure of biodiversity when predicting impacts and setting conservation targets.

## Detailed methods

### Environmental heterogeneity and spatial structure

Environmental heterogeneity (Fig 1A) was modelled by assigning to each site $x$ a set of $K$ independent random variables $E_{kx}$ ($1 \leq k \leq K$), representing, e.g. the principal component(s) of abiotic environmental variation. The $E_{kx}$ were sampled from spatially correlated Gaussian random fields [33] with mean 0, standard deviation 1, and correlation length 1.

Each species $i$ in the model was allocated environmental optima $E_{ik}^{\mathrm{opt}}$ for each environmental variable $k$, sampled from uniform distributions in the range $[1.25 \cdot \min_x E_{xk}; 1.25 \cdot \max_x E_{xk}]$. The effective heterogeneity of the environment was controlled by varying the niche width parameter $w$. The intrinsic growth rate of the $i$th species in site $x$ was computed as

$$R_{ix} = 1 - \left(\frac{2}{w}\right)^2 \sum_{k=1}^{K} (E_{kx} - E_{ik}^{\mathrm{opt}})^2, \tag{1}$$

that is, to simplify parameterisation, we assumed identical niche widths for all species and all $n$ independent components of environmental variation. For the random metacommunities we set $n$ to either 1 or 2 and observed no major change in outcomes. For the fitted metacommunites, the first two principal components of the observed environmental variation were used.

The spatial network of sites (Fig 1A) was modelled by randomly sampling Cartesian coordinates from a uniform distribution in the range $[0; \sqrt{N}]$, where $N$ is the number of sites. As in previous studies [17, 18], we linked nearby sites using a Gabriel graph [34].

## Scaling local site area

An essential technical innovation that made this study possible is a technique we developed to model local differences in site area by scaling the intensity of local ecological interactions. The precise functional form of the area dependence of intra- and inter-specific ecological interactions is not currently known. In order to overcome this knowledge gap we assembled metacommunities of different total area $a$ and measured the decay in effective interaction strengths. Previous work has shown that regional scale interaction coefficients $C_{ij}$ describing the interaction between species $i$ and $j$ can be estimated for models by perturbing the biomass of each species in turn and summing the impacts on all other species over the metacommunity ([17] and Section A in S1 Text). The resulting interaction matrix $\mathbf{C}$ captures the combined effects of differences in environmental preference, limited dispersal, indirect and direct interactions.

The decay in average interaction strength with area, which we refer to as the *competition area relation* (CAR), can be modelled by two power laws, one for inter-specific interactions, $\overline{C_{ij}} \propto a^v$ ($i \neq j$), and one for intra-specific interactions, $\overline{C_{ii}} \propto a^{v'}$ (Fig Aa in S1 Text). The exponents $v$ and $v'$ depend on model parameters, in particular the abiotic niche width $w$, which strongly affects species' range sizes (we found the effect of dispersal length to be weak by comparison). The CAR can be incorporated into the model dynamics in order to model variation in the local biomass (area) between sites by scaling the site-specific interaction matrix $\mathbf{A}_x$. Thus we model each site as an implicit sub-network of various nodes and scale the interactions to those expected for the corresponding (sub-)metacommunity.

Formally, if $\mathbf{A}_0$ is a hollow matrix with zeros on the diagonal, the scaled matrix is given by

$$\mathbf{A}_x = \lambda'_x \mathbf{I} + \lambda_x \mathbf{A}_0. \tag{2}$$

where $\mathbf{I}$ is the identity matrix, and

$$\lambda_x = \left(\frac{a_x}{a_0}\right)^v, \tag{3a}$$

$$\lambda'_x = \left(\frac{a_x}{a_0}\right)^{v'}. \tag{3b}$$

Here $a_x$ is the area of the $x$th site and $a_0$ the area of a reference site with (unscaled) interaction matrix $\mathbf{A} = \mathbf{I} + \mathbf{A}_0$. We measure site area in units such that $a_0 = 1$. For random metacommunities, the $a_x$ values were randomly assigned to sites $x$ such that they covered the range from 1 to 30 biomass units in equal intervals. For fitted metacommunities, the $a_x$ were extracted from the empirical species by site tables (see below). The exponents $v$ and $v'$ were set based on the relationships between the CAR and SAR found in simulations, assuming that within-site SARs are typically linear (see Section A in S1 Text).

## Metacommunity dynamics

Metacommunity dynamics were modelled using a spatial extension to the classic Lotka-Volterra community model [17, 18]

$$\frac{dB_{ix}}{dt} = \left( R_{ix} - \lambda'_x B_{ix} - \sum_{j=1}^{S} \lambda_x A_{ij} B_{jx} \right) B_{ix} + \sum_{y=1}^{N} B_{iy} D_{xy}, \tag{4}$$

where $B_{ix}$ represents the biomass of the $i$th species in the $x$th site. $R_{ix}$ are the intrinsic growth rates, which vary across the landscape.

For simplicity, the unscaled local interspecific interaction coefficients $A_{ij}$ were sampled from a discrete distribution with $P(A_{ij} = 0.3) = 0.3$, and $A_{ij} = 0$ otherwise. $D_{xy}$ are the elements of the spatial connectivity matrix describing the inter-site dispersal. Emigration and immigration rates are given by $D_{xx} = -e$, $D_{xy} = (e/k_y) \exp(-d_{xy}/\ell)$ for sites $x$, $y$ connected by the spatial network, and $D_{xy} = 0$ otherwise; $e$ is an emigration rate, $d_{xy}$ the distance between sites $x$, $y$, and $\ell$ the characteristic dispersal length, which was systematically varied. The normalisation $k_y$ represents the unweighted degree of the $y$th site.

## Metacommunity assembly and removal experiments

Model metacommunities were assembled by iteratively generating species $i$ with randomly sampled $E_{ik}^{\text{opt}}$ and $A_{ij}$, and introducing them as invaders to the model at low biomass. Dynamics were then simulated over 500 unit times using Eq 4. Before each invasion, the metacommunity was scanned for any species whose biomass had fallen below the extinction threshold of $10^{-4}$ biomass units in all sites. These species were considered regionally extinct and removed from the model. Metacommunity models assembled in this fashion eventually saturated with respect to both average local and regional species richness [17] due to the onset of ecological structural instability [35]. After saturation is reached, each invasion generates, on average, a single extinction.

Following pilot studies metacommunities were assembled with $w$ assigned 10 values logarithmically spaced in the range $0.5 \leq w \leq 15$. The parameter $\ell$ was logarithmically spaced in the range $2 \cdot 10^{-2} \leq w \leq 10$, again with 10 distinct values included. In both cases, a couple of extreme values were excluded from the analysis either because beyond a threshold, no further change in outcomes was observed or, in the case of $\ell$, because very small values lead to numerical errors in the simulation.

After assembling model metacommunities of 20 sites to regional diversity limits, each site in turn, and all associated edges were systematically removed and the simulation advanced over $10^4$ unit times. Regional diversity was then assessed by reference to the same extinction threshold and compared to pre-disturbance levels. For completeness, the dispersal matrix $\mathbf{D}$ was updated to reflect the new degree distributions. More complex removal experiments including multiple sites, perhaps removed sequentially, could be an interesting extension to the methodology employed here, however the combination of sites removed or the temporal sequence would greatly add to an already highly complex biotic response. Therefore we limit removals to single sites for simplicity.

## Supporting information

**S1 Text. Supporting information for "Community composition exceeds area as a predictor of long-term conservation value".**
(PDF)

## Author Contributions

**Conceptualization:** Jacob D. O'Sullivan, Axel G. Rossberg.

**Data curation:** Jacob D. O'Sullivan, Ramesh Wilson.

**Formal analysis:** Jacob D. O'Sullivan, Axel G. Rossberg.

**Funding acquisition:** Axel G. Rossberg.

**Investigation:** Jacob D. O'Sullivan.

**Methodology:** Jacob D. O'Sullivan, Axel G. Rossberg.

**Project administration:** Jacob D. O'Sullivan, Axel G. Rossberg.

**Resources:** Axel G. Rossberg.

**Software:** Jacob D. O'Sullivan.

**Supervision:** Axel G. Rossberg.

**Validation:** Jacob D. O'Sullivan.

**Visualization:** Jacob D. O'Sullivan.

**Writing – original draft:** Jacob D. O'Sullivan.

**Writing – review & editing:** Jacob D. O'Sullivan, J. Christopher D. Terry, Axel G. Rossberg.

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
