## [Decision Letter · Decision Letter 0]

18 Aug 2022

Dear Dr O'Sullivan,

Thank you very much for submitting your manuscript "Community composition exceeds area as a predictor of long-term conservation value" for consideration at PLOS Computational Biology. As with all papers reviewed by the journal, your manuscript was reviewed by members of the editorial board and by several independent reviewers. The reviewers appreciated the attention to an important topic. Based on the reviews, we are likely to accept this manuscript for publication, providing that you modify the manuscript according to the review recommendations.

Both reviewers' comments focus largely on clarity of the model components, computational methods, and conclusions, as well as citing and putting the current work in context with the appropriate literature. Please ensure that all parameters and terms used are clearly defined throughout, in addition to addressing the more specific reviewer comments.

Sincerely,

Nic Vega, Ph.D.

Academic Editor

PLOS Computational Biology

James O'Dwyer

Section Editor

PLOS Computational Biology

[LINK]

Both reviewers' comments focus largely on clarity of the model components, computational methods, and conclusions. Please ensure that all parameters and terms used are clearly defined throughout, in addition to addressing the more specific reviewer comments.

Reviewer's Responses to Questions

**Comments to the Authors:**

Reviewer #1: General comments

This is an interesting and well written paper examining the importance of community assemblage uniqueness for retaining biodiversity, through a metacommunity modelling approach. The key refinements required before publication are making the text and figures a bit more clear and self explanatory, particularly more clearly defining and describing equation terms that are used.

Specific comments

L50-64: This approach to metacommunity modelling (based on Lotka Voltera) has been used by quite a few others. You should cite some of them here. This is done in the methods section, but citations are also needed here.

L200: is it a proportion or a percentage? It's described as a proportion, but given here as a percentage.

Fig. 1: Use of terms in the figure image and legend should be defined in the legend. (Same goes for the other figures)

Fig 4: define 'w' in the legend

Fig. 5: Define the terms in the legend

Fig. 6: define what D and L represent in the legend; Add or describe units for community distinctness; replace 'both panels' with 'all panels' (there are four panels)

Reviewer #2: This study quantifies the relative importance of losing habitat sites and the associated uniqueness of species composition therein to the loss of species. This quantification is developed from a Lotka-Volterra metacommunity model that links multiple habitats through dispersal. Two empirical data sets are also used to "test" the simulation finding that the uniqueness of species composition (beta diversity) plays a greatly more important role than area per se in species extinctions. While the study is in general well presented and executed, I do have a number of questions that require clarifications.

Let me start with the empirical data first. Although the empirical data support the result that beta diversity is more important than area in determining extinction, I do not see this result requires the fitting of the LVMCM to the empirical data. In another words, I don't see the connection between panel A and panel B in Figure 6 and the connection is necessary. There is no need to do any "simulated removal experiment" for the two data sets. You can just randomly remove the empirical habitat sites (and the beta diversity) to generate panel B. I understand your interest to ensure that the parameter ranges of the theoretical model are indeed not far from those of empirical data but that is a different issue. The presentation of Figure 6 leaves an impression the results in panel B are from the fitted LVMCM. That is not true and more importantly not necessary. Need to clarify this point.

Following up the same thread and is also recognized in the paper, the fitting of the theoretical model to the data is not a traditional way of model fitting. It is a kind of error and trial approach to find a niche width to match the empirical OFD to the theoretical one. This is fine but we need to recognize that OFD is only one of the many patterns the LVMCM can generate. Many other mechanisms rather LV model can also generate a similar OFD. As such, the comparison between empirical and theoretical OFDs is a weak (if at all reasonable) test of the theoretical model. This problem should be recognized and discussed in the paper. Further to that, the study reads as if the species niche width w were solely determined by the mixing rate m. I believe this is not the case. I don't think it is ever possible to estimate niche width from an OFD at all. Anyway, what is really the mixing rate? The paper is not clear about that. Without a clear understanding of what is the mixing rate, it is not possible to interpret w that is so estimated.

Having a closer look at panel B of Figure 6, I was also surprised to see the result of lichen-fungi in the left panel. I appreciate authors' interpretation of the two two unexpected small sites. But they have not yet answered why these two sites have such high proportions of extinctions. The only possibility for this to occur is that the two small sites have a large number of endemic species, more than those in large sites. The negative correlation between species richness and environmental rarity obviously cannot explain this unexpected result. Anyway, I don't see the interpretation on page 8 is sufficient. My feeling is that the lichen-fungi data are not typical of empirical data.

That species composition uniqueness has a lot to do with species loss is not new but has been widely recognized in the literature, though the relative importance of area and composition uniqueness is less heeded and I appreciate the authors address this important question. However, I am still trying to wrap up my mind why area is so little important. It is a kind of hard to understand that because sites are where species (regardless of endemics or generalists) reside. When an area is lost, the endemics must lose. In this case, area should have similar, if not more, predictive power than composition uniqueness. These are the imminent extinctions. So, here the question comes to the extinction debt. Why area matters so little to the debt compared to beta diversity? It would be helpful to have some discussion on that.

Anyway, how "long-term" extinction is defined? extinctions at the new equilibrium of LVMCM or after a certain time steps? I may miss it but did not see that was mentioned in the paper. I would like to see some discussion how removal of area (biomass) precipitates the "secondary extinctions" through the network of metacommunity. Are they caused by the increase of competition pressure or reducing population size (therefore no longer viable)? How the intrinsic growth rates and/or the competition coefficients of the LVMCM change after habitat sites are destroyed?

I am not surprised to see that dispersal length is not important (page 6, line 122) because species (spatial) distribution in your model is ultimately determined by the niche selection formulation regardless of their dispersal. Only for those species that have wide niche width but small dispersal distance can dispersal leaves a dent. Otherwise, they don't.

Line 79-80: It might be beneficial to log-transform biomass and extinctions in the linear model given their allometric relationship although I don't think that will change the result.

The current study only presents extinctions up to losing one habitat site. I would like to see the full extinction trajectory from losing one site up to losing all sites of the metacommunity. This is really what happens in reality - many habitats can be lost in the real world. Of course, if all sites are destroyed, the extinction trajectory will merge regardless of whether biomass or beta diversity is used as the x-axis and the extinction curves should have a similar shape.

A few typos:

line 72: We first we asked...

line 200: 0.15% should read as 0.15?

line 306: "spaced in the range 2 should be ranged 3? (from -2 to 1).

Figure 5: I had hard time to understand the label of regression coefficient. Should not biomass (area) and BC beta each has a coef across the x-axis? Which is which in the figure?

Fangliang He

**Have the authors made all data and (if applicable) computational code underlying the findings in their manuscript fully available?**

Reviewer #1: Yes

Reviewer #2: **No: **Yes, the program is included but I don't see the two sets of empirical data used in the study.

PLOS authors have the option to publish the peer review history of their article (what does this mean?). If published, this will include your full peer review and any attached files.

Reviewer #1: No

Reviewer #2: **Yes: **Fangliang He

Figure Files:

Data Requirements:

Reproducibility:

References:

---

## [Editor Report · Decision Letter 1]

9 Dec 2022

Dear Dr O'Sullivan,

We are pleased to inform you that your manuscript 'Community composition exceeds area as a predictor of long-term conservation value' has been provisionally accepted for publication in PLOS Computational Biology.

Best regards,

Nic Vega, Ph.D.

Academic Editor

PLOS Computational Biology

James O'Dwyer

Section Editor

PLOS Computational Biology

In their response to reviewers, the authors have sufficiently addressed the remaining concerns for this manuscript.

---

## [Editor Report · Acceptance letter]

18 Jan 2023

PCOMPBIOL-D-22-00580R1 

Community composition exceeds area as a predictor of long-term conservation value

Dear Dr O'Sullivan,

I am pleased to inform you that your manuscript has been formally accepted for publication in PLOS Computational Biology. Your manuscript is now with our production department and you will be notified of the publication date in due course.

With kind regards,

Anita Estes
